# Antioxidant, Antimutagenic and Cytoprotective Properties of Hydrosos Pistachio Nuts

**DOI:** 10.3390/molecules24234362

**Published:** 2019-11-29

**Authors:** Luis Noguera-Artiaga, Joel Said García-Romo, Ema C. Rosas-Burgos, Francisco Javier Cinco-Moroyoqui, Reyna Luz Vidal-Quintanar, Ángel Antonio Carbonell-Barrachina, Armando Burgos-Hernández

**Affiliations:** 1Departamento de Tecnología Agroalimentaria, Grupo Calidad y Seguridad Alimentaria (CSA), Escuela Politécnica Superior de Orihuela (EPSO), Universidad Miguel Hernández de Elche (UMH), Carretera de Beniel, km 3,2. 03312-Orihuela, Alicante, Spain; lnoguera@umh.es (L.N.-A.); angel.carbonell@umh.es (Á.A.C.-B.); 2Departamento de Investigación y Posgrado en Alimentos, Universidad de Sonora, Apartado Postal 1658, Hermosillo, Sonora 83000, Mexico; joelsa.garciaro@gmail.com (J.S.G.-R.); carina.rosas@unison.mx (E.C.R.-B.); javier.cinco@unison.mx (F.J.C.-M.); reynaluz.vidal@unison.mx (R.L.V.-Q.)

**Keywords:** Ames test, cancer, rootstock, MTT assay, *Pistacia vera*

## Abstract

Pistachio nuts are included among the foods with the highest antioxidant capacity. Stressed cultivating conditions, such as the use of regulated deficit irrigation (RDI), are expected to create a plant response that might increase the production of secondary metabolites. Fruits that are obtained under RDI treatments are commonly called hydroSOS products. The aim of this work was to study the influence of using different rootstocks (*P. atlantica*, *P. integerrima*, and *P. terebinthus*) and two RDI treatments on the antioxidant (ABTS, ferric reducing antioxidant power (FRAP), and DPPH), antimutagenic (Ames test), and cytotoxicity (MTT assay in five human cell lines) activities of pistachios. *P. terebinthus* showed the best antioxidant activity, and the RDI treatments maintained and improved the antioxidant properties of pistachios. Neither the rootstock nor the RDI had significant impact on the antimutagenic potential of pistachios. The nut extracts had no toxic effect on non-cancerous cells and the application of RDI did not reduce their cytoprotective capacity. Furthermore, neither rootstock nor RDI treatments affected the ability of the pistachio extracts of preventing the oxidative damage by H_2_O_2_. The application of RDI strategies, in addition to allowing irrigation water saving, led to obtaining pistachios with the same or even better biofunctional characteristics as compared to fully irrigated pistachios.

## 1. Introduction

The genus *Pistacia* belongs to the *Anacardiaceae*, which is a family that comprises about 70 genera and over 600 species. *Pistacia vera* is the only species of the genus commercially cultivated, and the rest of the species are mostly used as rootstocks [1]. As pistachio cultivation requires the use of rootstock, it is essential to study its influence on nut quality and functional activity. The main pistachio rootstocks are (i) *P. terebinthus* L. in the Mediterranean basin, (ii) *P. integerrima* L., and (iii) a hybrid between *P. integerrima* Steward ex Brandis and *P. atlantica* Desf. (UCB-I) in the USA, and *P. vera* L. in Iran [2,3].

Water is a scarce commodity and all productive sectors depend on it, especially agriculture. Therefore, it is necessary to improve the efficiency of its use. Regulated deficit irrigation (RDI) is a system of managing water supply by imposing different levels of water deficit at specific phenological stages, which have been found to be less sensitive, with no or low yield reduction and, consequently, low or no loss of economic benefits to farmers [4]. Vegetables and fruit that are cultivated under RDI are marketed under the brand “hydroSOStainable or hydroSOS” products. This food category is characterized by its environmental respect (optimized use of irrigation water) and a theoretical increase of secondary metabolites, which will improve the functionality and quality [5]. To be capable of categorizing a product as hydroSOS, it is necessary that it meets strict water control requirements during cultivation. This characterization is based on 16 hydric indicators (each one providing different marks or scores), and their sum allows for classifying orchard management as a hydroSOS one or not [6].

The application of RDI depends on the phenological phases of each crop. In the case of pistachio, three different phases can be easily distinguished during fruit development: (i) the nut grows up to its maximum size, (ii) the shell hardens, and (iii) the growth of the edible part occurs. The second stage is the one at which the pistachio nut is less sensitive to water stress; thus, strategies that aimed at reducing water supply during the cultivation of this nut must be applied at this specific time period.

Cancer is one of the leading causes of death in the world. Environmental factors, such as carcinogens, viruses, chemicals, and radiation, as well as by a genetic history, cause it [7]. Exposure to mutagens is one of the main causes of cancer development, causing the mutation of genes that are directly involved in the regulation of the cell cycle. Lung, colorectal, stomach, liver, and breast are the five types of cancer that cause the highest number of deaths [8]. Many natural products have the potential to trigger apoptosis in numerous human cancer cell types. Nowadays, a number of plant-based agents are clinically used as a support therapy in cancer treatment, proving the goodness of these type of products [9]. Hence, it is necessary to search for new plant-derived products as apoptosis inducers [10,11].

Pistachio nuts have been recently ranked among the first 50 highest antioxidant food products and they a rich source of phenolic compounds [12]. Pistachio contains epicatechin, quercetin, kaempferol, cyaniding-3-*O*-galactoside, cyanindin-3-*O*-glucoside, among other polyphenols [13]. It has been shown that pistachio polyphenols are bioaccesible during simulated human digestion, releasing more than 90% of its total content in the gastric compartment [14]. Consequently, pistachios have a high number of bioactive compounds, and the human body can assimilate and use these.

The main aims of this work were (i) to study the antioxidant, antimutagenic, and cytoprotective properties of pistachios and (ii) the influence of the rootstock (n = 3) and a decrease in the application of irrigation water during cultivation on their main functional activity.

## 2. Results and Discussion

### 2.1. Antioxidant Tests 

The ABTS^+^ method, which is based on the capability of presumptive antioxidant to reduce the ABTS^+•^ radical, is one of the most widely used methods, because it uses a stable reaction, is practical, highly sensitive, and fast to carried out. Additionally, this method allows confirmation of antiradical capacity of either hydrophilic and lipophilic antioxidants [15]. The free radical scavenging capacities of ABTS^+•^ showed that pistachios that were using the *P. terebinthus* rootstock had the highest antioxidant activity (3.67 mM TE g^−1^), followed by *P. integerrima* (3.23 mM TE g^−1^), and finally *P. atlantica* (2.80 mM TE g^−1^) (Table 1). In the case of pistachios that were obtained under different irrigation treatments, nuts from T2-trees had a higher antioxidant activity than samples from trees that were subjected to the other two irrigation treatments (T0 and T1), with an antioxidant activity of 3.92 mM TE g^−1^ versus 2.86 and 2.96 mM TE g^−1^, respectively (Table 1). In a study evaluating key characteristics of eight pistachio cultivars by Noguera-Artiaga [16], similar values of antioxidant activity for the Kerman cultivar were found, the same cultivar used in the current experiment.

The antioxidant activity of the hydrophilic fraction of pistachios was measured while using the ferric reducing antioxidant power (FRAP) method, which is based on the ability of an antioxidant to reduce Fe^+3^ in the presence of 2,4,6-Tripyridyl-s-triazine (TPTZ), forming a Fe^+2^-TPTZ complex [17]. After the analysis of samples that were obtained while using different rootstocks (Table 1), samples *P. integerrima* and *P. terebinthus* had the highest antioxidant activity with values of 1.37 and 1.32 mM TE g^−1^, respectively, while the nuts from *P. atlantica* had 1.23 mM TE g^−1^. Taking the irrigation treatment factor into account, the T1-sample had the highest antioxidant activity, with 1.37 mM TE g^−1^, followed by a group consisting of the samples T0 and T2 (1.29 mM TE g^−1^). Taghizadeh [18] obtained similar results after analyzing the antioxidant activity by the FRAP method of pistachio extracted while using different solvents.

As shown in Table 1, statistical differences were also found after the analysis of the antioxidant activity of pistachio extracts while using the DPPH^•^ method. Pistachios that were obtained on *P. terebinthus* rootstock showed the highest antioxidant activity (5.63 mM TE g^−1^), while the *P. integerrima* samples had the lowest value (5.07 mM TE g^−1^). Regarding the irrigation treatments, the DPPH method was the most effective one in differentiating among samples. The antioxidant activity increased with water restriction (deficit irrigation); the control sample (T0) showed the lowest antioxidant activity (4.59 mM TE g^−1^), followed by T1 samples (5.07 mM TE g^−1^), and finally T2 samples had the highest value (6.41 mM TE g^−1^).

In view of above described results, it can be concluded that the use of the *P. terebinthus* rootstock led to the highest antioxidant activity (according to the three methods used). This can be associated to the fact that *P. terebinthus* rootstock is, amongst the three rootstocks studied, the one with the highest polyphenolic content and highest amount of betulinic acid, according to Noguera-Artiaga [13]. Besides, it can also be concluded that deficit irrigation treatments (T1 and especially T2) maintained or improved the antioxidant (hydrophilic) capacity of pistachios. In plants that were cultivated at water deficit conditions, an excess of CO_2_ assimilated could have increased the biosynthesis of carbon-based secondary metabolites when carbohydrates exceed the amount used for growth concentrations [19].

### 2.2. Antimutagenicity Test

The number of histidine^+^ revertant colonies obtained was 120 ± 5 (spontaneous revertants), while, in the presence of the mutagen (sodium azide, 30 µg plate^−1^), 1652 ± 33 revertants plate^−1^ were counted. Samples inhibited an average of 8% of the mutagenicity (92% revertants) induced by SA, without significant differences with control (Figure 1). No differences were observed for irrigation treatment and rootstock. The pistachio extracts were neither toxic nor mutagenic to the bacteria at the tested concentrations, and bacterial growth was normal. Rajaei [20] obtained similar results when the antimutagenic activity of pistachio green hull extract was assayed, since they found that hydrophilic pistachio extracts (raw, unpurified) did not show antimutagenic activity, although, in this case the study was carried out while using 2-nitrofluorene as a mutagen control instead of sodium azide.

These results showed that the phenolic compounds of pistachio extracts had no activity against sodium azide. The application of different irrigation treatments or different rootstocks had no significant impact on the antimutagenic potential of pistachios.

### 2.3. Cytotoxic Test

*In vitro* cytotoxicity assessment is being increasingly recognized as an effective indicator of the toxic potential of compounds against cancerous cells. The MTT test assesses cellular viability measuring intracellular reduction of MTT reagent (water soluble and yellow) to a formazan salt (water insoluble and purple), which can be colorimetrically detected [21].

The effect that regulated deficit irrigation and rootstocks may have had on the functional properties of pistachios was studied by evaluating the cytotoxic effect of pistachio extracts on four cancerous and one non-cancerous cell lines (Table 2). The results obtained from testing samples were compared to those that were obtained with cisplatin, a positive cytotoxic control agent, which is a genotoxic drug that is used in chemotherapy. In two of the cancerous cell lines under study (HCT-116 and MDA-MB-231), no statistically differences were found to be associated to neither rootstock nor irrigation treatment factors. In the case of the HCT-116 cell line, the pistachio samples analyzed caused an inhibition of ~60% (~40% viability), achieving the same effect as a cisplatin dose that 50 μg mL^−1^ did. The same effect was observed in the MDA-MB-231 cell line with a cellular viability of ~60% (Table 2).

In A549 and HeLa cell lines, no statistically differences were observed among the samples from different irrigation treatments (Table 2). However, rootstock had significant differences when compared to the control (cisplatin, 50 mg mL^−1^), with pistachio samples having statistically higher viability (average ~68 and ~61% of viability in A549 and HeLa lines, against ~61 and ~57%, respectively) than when cisplatin was used.

In the ARPE-19 non-cancerous cell line, no differences were found among the samples that were obtained under different irrigation treatments, but significant differences were found when the rootstock factor was analyzed (Table 2); the *P. atlantica* rootstock led to the highest cellular viability (94%). In addition, the three rootstock samples had higher cellular viability than that observed for the control treatment. An important issue to point out is that none of the pistachio extracts reduced cellular viability of ARPE-19 below 85%; thus, they cannot be considered to be cytotoxic up to 100 μg mL^−1^.

These results showed that the compounds that were present in pistachio extracts had no toxic effect on ARPE-19 non-cancerous cells. On the contrary, they could significantly affect all four human cancer lines, with the A549 cell line being the most resistant and HeLa the most sensitive. Similar results have been previously reported by Seifaddinipour [22] on extracts from pistachio hulls on six cancerous cell lines and normal fibroblast, in which the same methodology used in the present study was followed. On the other hand, the application of the regulated deficit irrigation treatments had not diminished the original cytotoxic capacity that control pistachios have on the carcinoma cell lines studied.

It has been shown in other studies [13] that neither regulated deficit irrigation nor the use of different rootstocks affected the total concentration of triterpenoids. Pistachios are characterized by having triterpenoids, especially betulinic, oleanolic, and ursolic acids. Several studies have established relationships between phenolic compounds and their activity against tumor cell lines, as tested in vitro [23,24].

### 2.4. Cellular Viability and Oxidative Damage Tests

The lowest H_2_O_2_ concentrations (0.062 and 0.125 mM) tested had no significant effect on the ARPE-19 cells as compared to the control (0 mM of H_2_O_2_), obtaining values that were close to 100% cellular viability (Figure 2). Subsequently, a significant negative relationship between the H_2_O_2_ concentration and cellular viability was observed (R^2^ = 0.85), with the viability reaching values of ~10%. Based on these results, the concentration of 10 mM of H_2_O_2_ was chosen for performing the cellular damage test. This selection was based on the fact that this concentration led to a cellular viability close to 20%, which caused obvious damage to the cells, but allowed enough cell survival to guarantee that a test would be performed at optimal conditions for its full development.

Figure 3 shows the results that were obtained after incubation of ARPE-19 cells during 4 and 24 h with the pistachio extracts. As previously discussed, when 10 mM of H_2_O_2_ was used, cellular viabilities of ~20% were obtained, without having significant differences among the studied samples. No statistically differences were found for the irrigation treatments when the cells were incubated with the pistachio extracts, leading to ~95% cellular viability. However, the *P. atlantica* rootstock led to a higher cellular viability (~100%) as ~95% for *P. integerrima* and *P. terebinthus*. When H_2_O_2_ (10 mM) was added to the cells for 30 min. (after 4 h of incubation with the pistachio extracts), a viability of ~30% was obtained in all of the samples. This suggests that the compounds that were present in pistachio extracts were able to reduce the oxidative damage that was observed in ARPE-19 cells by approximately 10% after 4 h (Figure 3A).

The results were similar to those obtained after 4 h of incubation by repeating the same process but incubating ARPE-19 cells with the pistachio extracts for a longer time, 24 h. There were no significant differences among irrigation treatments but *P. atlantica* showed a better behavior (~95% cellular viability) as compared to ~85% for *P. integerrima* and *P. terebinthus* (Figure 3B). When H_2_O_2_ (10 mM) was added to the cells for 30 min. (after 24 h of incubation with the pistachio extracts), a viability of ~55% was obtained in all of the samples (without statistically differences among them). In this case, the pistachio extracts were able to protect ARPE-19 cells from oxidative damage by approximately 35%. The cellular internalization depends on exposure time of cell to pistachio extracts. Increasing the exposure time increases the amount of pistachio compounds into the cell, so that greater protection against oxidative stress caused by peroxide radicals (formed from H_2_O_2_ by Fenton or Haber-Weiss reactions) was achieved.

Based on the above, neither rootstock nor regulated deficit irrigations significantly affected the original protective level against oxidative damage by H_2_O_2_ of the control pistachio extracts. On the other hand, it is necessary to mention that the effect of protection against oxidative stress that is caused by hydrogen peroxide does not always have a linear behavior and, therefore, it is necessary to carry out further research in this regard [25].

## 3. Materials and Methods

### 3.1. Plant Material and Experimental Design

The experiment was conducted on the experimental farm “El Chaparrillo”, Ciudad Real, Spain (L 3°56′ W; L 39°0′ N; altitude 640 m) during the crop season 2016–2017. The plant material consisted of pistachio trees *P. vera* L. cultivar Kerman was budded on three rootstocks (i) *P. terebinthus* L., (ii) *P. atlantica* Desf., and (iii) *P. integerrima* L. The tree-spacing was set at 7 × 6 m (238 trees ha^−1^). Peter cv. was used as male tree and it was evenly distributed throughout the field, in a proportion of 10%. The climate of the area had an average annual rainfall of 397 mm, mostly distributed outside a four-month summer drought period, and the surface is a shallow 1.3 m deep clay-loam (Alfisol Xeralf Petrocalcic Palexeralfs) soil, and a discontinuous petrocalcic horizon of around 0.75 m. The orchard was managed under no tillage conditions; the weeds were controlled with post-emergence herbicides. Pest control and fertilization practices were those usually followed by local growers.

The control plants (T0) were irrigated at 100% of crop irrigation requirements (ETc) of the previous week, according to daily reference evapotranspiration (ETo), a crop factor, and while taking the canopy size into consideration [26]. In addition to T0, two RDI treatments (T1 and T2) were applied during stage II (fruit growth), the non-critical period. In these treatments, the water deficit was increased and the threshold values were −1.5 MPa (T1) and −2.0 MPa (T2) [4].

### 3.2. Extraction of Functional Compounds

The preparation of pistachio extracts was conducted, as previously described by Noguera-Artiaga [13]. One gram of grounded pistachio was weighed and combined with 5 mL of hexane and 8 mL of 30% of aqueous methanol (*v*/*v*), amended with 1% of ascorbic acid. The suspension was stirred, sonicated during 15 min. in an ultrasonic bath (JP Selecta S.A, model 3000512 Barcelona, Spain) with constant frequency (40 kHz), left overnight at room temperature at darkness, and then centrifuged for 10 min. (20.878× *g* at 4 °C). The hydrophilic phase was separated, dried under N_2_ stream, re-dissolved with sterile DMSO (solution of 100 mg mL^−1^), and stored (−24 °C) until the analyses. This extraction methodology has been shown to be one of the most effective ones when it comes to extracting the greatest number of phenolic compounds [20]. A negative control (5 mL of hexane and 8 mL of 30% of aqueous methanol (*v*/*v*) amended with 1% of ascorbic acid) was used in all determinations in order to obtain reliable results. The results shown by this control were subtracted from the results that were obtained by the samples to obtain the final result.

### 3.3. Antioxidant Tests

Free radical scavenging capacities were determined by ABTS [27], DPPH [28], and FRAP [29] assays, with some modifications to be used in 96-well microplates.

The evaluation of antioxidant activity with ABTS reagent [2,2-Azino-bis(3-ethylbenzothiazoline-6-sulfonic acid)] was carried out according the method that was previously described by Loarca-Piña [30]. Briefly, 20 µL of pistachio extracts were mixed with 230 µL of ABTS solution, and, after reaction, absorbance was recorded at 734 nm in a Beckman Coulter AD 340 Microplate reader (Beckman Coulter, CA, USA).

A 150 µM DPPH solution was prepared in 80% methanol in order to perform the DPPH assay. A 200-μL aliquot of this solution was combined with 100 uL of pistachio extract, placed into each well, and the plate was covered and left at darkness at room temperature for 6 min. After reaction, absorbance was determined at 520 nm in a Beckman Coulter AD 340 Microplate reader [31].

The ferric reducing antioxidant power (FRAP) assay measures the capability of reducing ferric ions. FRAP reagent was freshly prepared by mixing 100 mL of a sodium acetate buffer (0.3 M, pH 3.6) solution, 10 mL of 2,4,6-tri(2-pyridyl)-1,3,5-triazine (TPTZ) solution (10 mM in 40 mM HCl), and 10 mL of FeCl_3_ 6H_2_O (20 mM) solution. A 200-μL aliquot of the pistachio extract was mixed with the FRAP solution for 30 min. at darkness to obtain a proper and even reaction. The absorbance of samples was recorded at 593 nm while using a Beckman Coulter AD 340 Microplate reader [32]. Trolox was used as a standard antioxidant. All of the analyses were run in triplicate and the results were expressed as mM Trolox equivalents g^−1^.

### 3.4. Antimutagenicity Test

The antimutagenic activity of pistachio extract was evaluated while using the standard mutagenicity assay, as described by Maron and Ames [33], with *Salmonella typhimurium* TA100 as a tester strains, in the presence of sodium azide (SA) as positive control. The pistachio extracts were diluted in sterile DMSO and spiked with sufficient SA to reach 20.0, 2.0, and 0.2 µg mL^−1^. All of the assays were performed in triplicate. Antimutagenic activity was reported as the percentage of SA-induced revertants per plate inhibited due to the presence of pistachio extract.

### 3.5. Cell Lines

Human cell lines A-549 (lung carcinoma), HeLa (epithelial cervix adenocarcinoma), MDA-MB-231 (breast adenocarcinoma), HCT 116 (colon carcinoma), and ARPE-19 (retinal non-cancerous epithelial) were used. The cell lines were obtained from the American Type Culture Collection (Rockville, MD). All cell lines were maintained in Dulbecco’s modified Eagle’s medium and RPMI-1640 Medium (Sigma Aldrich, St. Louis, MO, USA), supplemented with 10 and 15% heat-inactivated fetal bovine serum (FBS) (Gibco, Grand Island, NY, USA), and grown at 37 °C in an atmosphere of 5% CO_2_.

### 3.6. Cytotoxic Test

MTT assay (Roche, cell proliferation kit I, Roche, Cat. No. 11-465-007-001) was used to study the cytotoxic effect of pistachio extracts on human cancer cell lines. In a 96-flat-well plate, 1 × 10^4^ cells well^−1^ were seeded and suspended in 100 µL of medium. After 24 h of incubation at 37 °C and 5% CO_2_ atmosphere, the cell cultures were maintained for another 24 h at same conditions with the addition of 100 µL of culture medium without FBS and the pistachio extracts re-suspended in DMSO (0.5%) at final a concentration of 100 μg mL^−1^. The control cell cultures did not show any evidence of cell damage. Cisplatin [*cis*-Diamminedichloroplatinum (II)] was used as a positive control for cytotoxicity at a concentration of 50 µg mL^−1^ for all of the studied cell lines, except for MDA-MB-231, in which case the concentration was 100 µg mL^−1^. The plates were read the next day while using an ELISA plate reader (Benchmark Microplate Reader; Bio-Rad, Hercules, CA, USA) [34].

### 3.7. Cytoprotection Against H2O2-Induced Cell Damage Activity

A dose response curve to determine the cellular viability in the presence of H_2_O_2_ was prepared in order to find the H_2_O_2_ concentration to be used to cause oxidative cell damage. The ARPE-19 cells were seeded (1 × 10^4^ cells well^−1^) into 96-well plates, suspended in 100 µL of DMEM, and then incubated for 24 h at 37 °C and 5% CO_2_ atmosphere. Subsequently, cells were treated with different concentrations of H_2_O_2_ (0.062, 0.125, 0.250, 0.5, 1.0, 10.0, and 20.0 mM) for 30 min. After the treatment, the cell medium was replaced (by an equal volume of DMEM serum-free medium) and the original MTT protocol was followed.

The analysis of the protection against reactive oxygen species was conducted when the concentration of H_2_O_2_ to be used was known. ARPE-19 cells, 1 × 10^4^ cells well^−1^, were plated in 96-well plates for 24 h (suspended in 100 µL of DMEM, at temperature of 37 °C, and controlled atmosphere with 5% CO_2_). Afterwards, cells were incubated with extracts obtained from pistachios cultivated under different irrigation treatments and while using different rootstocks, at a concentration of 100 μg mL^−1^, during 4 and 24 h. After that, the medium was retrieved from wells and they were incubated with 10 mM of H_2_O_2_ for 30 min. (at same temperature and atmosphere). After treatment, an equal volume of DMEM serum-free medium replaced the cell medium and MTT assay was performed [35,36].

### 3.8. Statistical Analysis

The data that are presented in this study are the mean values of, at least, three replicates, and were subjected to three-way (in case of antimutagenicity test) and two-way (rest of determinations) analysis of variance (ANOVA). Subsequently, the data were subjected to Tukey’s multiple-range test to compare the means. Differences were considered to be statistically significant at *p* < 0.05. All of the statistical analyses were done while using XLSTAT software (ADDINSOFT XLSTAT version 2014.1, Paris, France).

## 4. Conclusions

The experimental results have demonstrated that the use of the *P. terebinthus* rootstock led to pistachios with higher antioxidant activity, while application of RDI strategies maintained or improved pistachios antioxidant capacity. Neither the use of different rootstocks nor the application of RDI during pistachio cultivation had a significant impact on the antimutagenic potential and cytoprotective activity of pistachios. Of the four cancer cell lines studied, A549 was the most resistant and the HeLa cell line was the most sensitive to pistachio extracts, which successfully protected ARPE-19 cells from oxidative damage that was caused by H_2_O_2_ at a level of 10% when incubated during 4 h, and at 35% after 24 h of exposure. Thus, the application of RDI strategies in pistachios allows for the saving of water during pistachio cultivation and led to producing pistachio nuts with the same or better functional characteristics than the control samples.

## Figures and Tables

**Figure 1 molecules-24-04362-f001:**
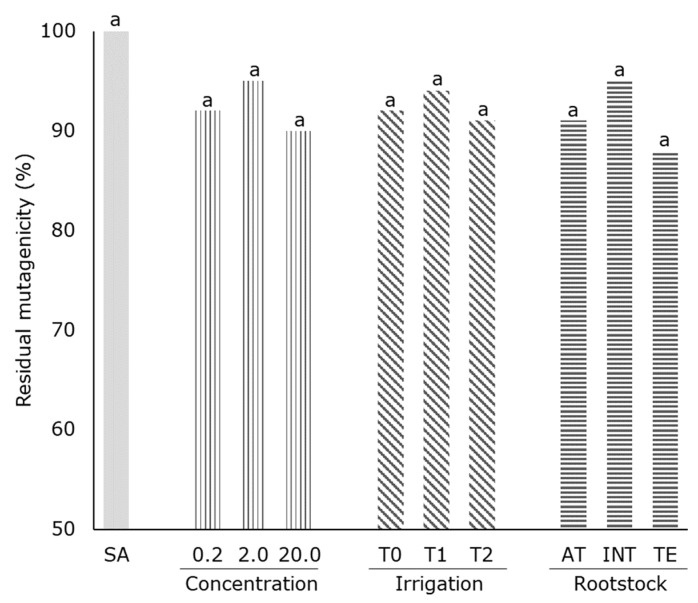
Antimutagenicity test of pistachio extracts (% inhibition of sodium azide (SA) mutation) at different concentrations (0.2, 2.0, and 20.0 mg mL^−1^). Values with same letters within a same factor were not significantly different (*p* < 0.005), Tukey’s least significant difference test. Spontaneous revertants 120 ± 5 and SA control (30 µg plate^−1^) induced 1652 ± 33 revertants plate^−1^.

**Figure 2 molecules-24-04362-f002:**
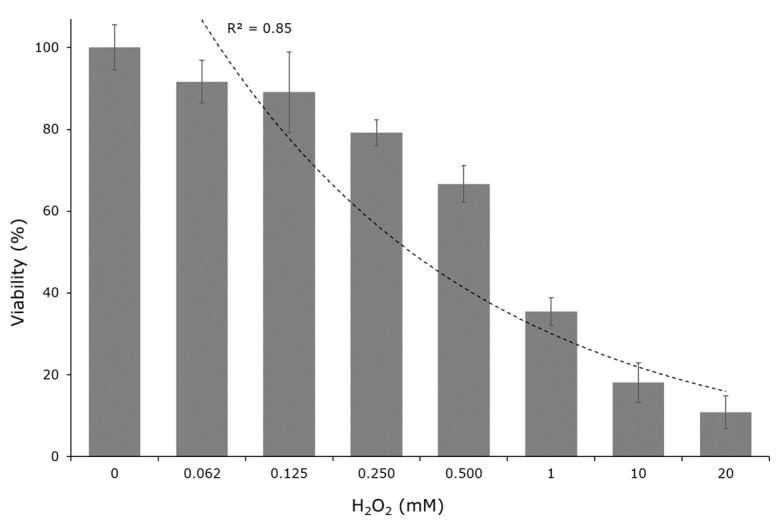
Cellular viability evaluated by MTT method in ARPE-19 cells exposed to different concentration of H_2_O_2_ during 30 min.

**Figure 3 molecules-24-04362-f003:**
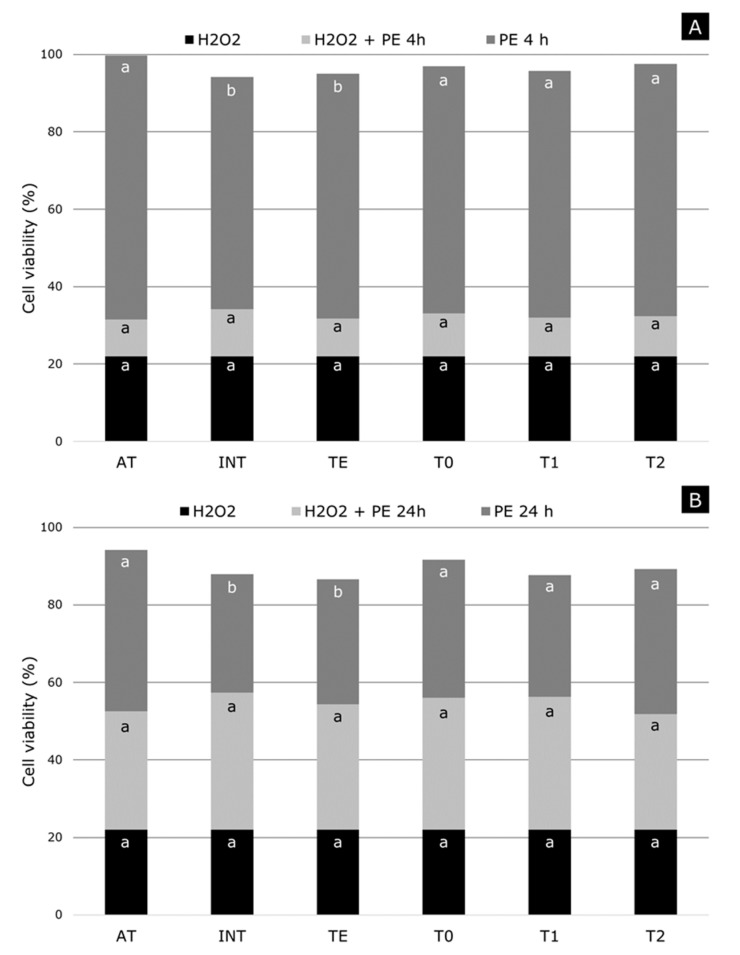
Cellular viability evaluated by MTT method of ARPE-19 cells treated with pistachio extracts (PE), as affected by rootstock and irrigation treatment, at 100 μg mL^−1^ after 4 h (**A**), and 24 h (**B**), and then exposed to 10 mM H_2_O_2_ during 30 min. Different letters within each factor means significant differences (*p* ≤ 0.05); Tukey’s least significant difference test.

**Table 1 molecules-24-04362-t001:** Antioxidant activity [mM Trolox equivalents (TE) g^−1^] of pistachio extracts, as affected by rootstock and irrigation treatment.

Figure	ABTS	FRAP	DPPH
	(mM TE g^−1^)		
	**ANOVA ^1^**		
**Rootstock**	***	***	***
**Irrigation**	***	**	***
**Rootstock × Irrigation**	***	**	**
	**Tukey multiple range test ^2^**		
**Rootstock**			
*P. atlantica*	2.80 c	1.23 b	5.44 ab
*P. integerrima*	3.23 b	1.37 a	5.07 b
*P. terebinthus*	3.67 a	1.32 a	5.63 a
Irrigation			
T0	2.86 b	1.29 b	4.59 c
T1	2.96 b	1.37 a	5.13 b
T2	3.92 a	1.29 b	6.41 a
**Rootstock × Irrigation**			
*P. atlantica* × T0	2.37 d	1.24 ab	4.83 d
*P. atlantica* × T1	2.61 d	1.34 a	6.07 b
*P. atlantica* × T2	3.41 b	1.10 b	5.42 c
*P. integerrima* × T0	2.95 cd	1.36 a	4.43 e
*P. integerrima* × T1	2.90 cd	1.41 a	4.42 e
*P. integerrima* × T2	3.94 ab	1.37 a	6.36 b
*P. terebinthus* × T0	3.24 bc	1.28 a	4.51 e
*P. terebinthus* × T1	3.36 bc	1.39 a	4.89 d
*P. terebinthus* × T2	4.40 a	1.26 a	7.47 a
**Pooled variance**	0.32	0.15	0.22

^1.^ NS: not significant at *p* < 0.05; **, and ***, significant at *p* < 0.01, and 0.001, respectively. ^2.^ Values (mean of three replications) followed by the same letter, within the same column and factor, were not significantly different (*p* < 0.05), Tukey’s least significant difference test.

**Table 2 molecules-24-04362-t002:** Viability (%) of pistachio extracts as affected by rootstock and irrigation treatment, at 100 μg mL^−1^ on human cancerous and non-cancerous cell lines.

Factor	HCT-116	A549	HeLa	MDA-MB-231	ARPE-19
	(% Viability)				
	**ANOVA ^1^**				
**Rootstock**	NS	***	**	NS	***
**Irrigation**	NS	NS	NS	NS	***
**Rootstock × Irrigation**	NS	***	***	NS	***
	**Tukey multiple range test ^2^**				
**Rootstock**					
*P. atlantica*	38.9	67.4 a	30.1 a	61.3	94.2 a
*P. integerrima*	39.0	68.3 a	30.1 a	61.9	87.9 b
*P. terebinthus*	43.4	68.7 a	32.3 a	60.3	86.6 b
CISP ^3^	40.2	61.1 b	28.2 b	57.2	69.2 c
**Irrigation**					
T0	39.7	65.3	28.3	58.0	91.7 a
T1	39.8	64.4	27.0	60.8	87.7 a
T2	40.9	61.1	28.2	61.0	89.3 a
CISP ^3^	40.2	61.1	28.2	57.2	69.2 b
**Rootstock × Irrigation**					
*P. atlantica* × T0	36.1	73.2 a	30.1 a	62.5	99.1 a
*P. atlantica* × T1	35.8	66.2 a	29.3 a	60.9	91.1 a
*P. atlantica* × T2	44.3	64.6 ab	30.4 a	59.6	90.9 a
*P. integerrima* × T0	36.3	74.5 a	28.4 ab	55.5	84.9 ab
*P. integerrima* × T1	41.0	67.2 a	31.5 a	64.6	88.3 ab
*P. integerrima* × T2	38.8	65.1 ab	29.6 a	64.9	90.6 a
*P. terebinthus* × T0	46.7	63.7 ab	34.8 a	58.9	90.1 a
*P. terebinthus* × T1	42.7	75.3 a	28.8 ab	59.8	83.8 ab
*P. terebinthus* × T2	39.9	69.1 a	33.3 a	61.3	86.2 ab
CISP ^3^	40.2	61.1 b	28.2 b	57.2	69.2 b
**Pooled variance**	4.1	2.3	1.8	4.8	4.5

^1.^ NS: not significant at *p* < 0.05; **, and ***, significant at *p* < 0.01, and 0.001, respectively.^2.^ Values (mean of 3 replications) followed by the same letter, within the same column and factor, were not significantly different (*p* < 0.05), Tukey’s least significant difference test. ^3.^ Cisplatin (CISP) concentration was 50 µg mL^−1^ for all the cell lines studied, except for MDA-MB-231 in which case the concentration was 100 µg mL^−1^.

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
