# Peer review of "Antioxidant, Antimutagenic and Cytoprotective Properties of Hydrosos Pistachio Nuts"

_molecules, 2019, doi:10.3390/molecules24234362_

Round 1
Reviewer 1 Report
Introduction line 62-63:
The AA stated: “Nowadays, a number of plant-based agents are clinically used in cancer treatment proving the goodness of these type of products”; better to use the term “support to anticancer therapy”, and to indicate more specific references, such as ACQUAVIVA R, SORRENTI V, SANTANGELO R, CARDILE V, TOMASELLO B, MALFA G, VANELLA L, AMODEO A, GENOVESE C, MASTROJENI S, PUGLIESE M, RAGUSA M, DI GIACOMO C (2016). EFFECTS OF AN EXTRACT OF CELTIS AETNENSIS (TORNAB.) STROBL TWIGS ON HUMAN COLON CANCER CELL CULTURES. ONCOLOGY REPORTS, 36(4):2298-2304. ISSN: 1791-2431, DOI: http://dx.doi.org/10.3892/or.2016.5035
Introduction line 70 :
Add : The activity is confirmed on extracts having the same chemical composition by others AA. ( insert references: ACQUAVIVA R, D’ANGELI F, MALFA GA, RONSISVALLE S, GAROZZO A, STIVALA A, RAGUSA S, NICOLOSI D, SALMERI M, GENOVESE C. (2019) ANTIBACTERIAL AND ANTI-BIOFILM ACTIVITIES OF WALNUT PELLICLE EXTRACT (JUGLANS REGIA L.) AGAINST COAGULASE-NEGATIVE STAPHYLOCOCCI. NATURAL PRODUCT RESEARCH, 1-6. DOI: 10.1080/14786419.2019.1650352
References
The works n. 25 and 26 are very old, replace with:
GENOVESE C., ACQUAVIVA R., RONSISVALLE S., TEMPERA G., MALFA G.A., D’ANGELI F., RAGUSA S., NICOLOSI D. (2019). IN VITRO EVALUATION OF BIOLOGICAL ACTIVITIES OF OROBANCHE CRENATA FORSSK. LEAVES EXTRACT. NATURAL PRODUCT RESEARCH, 1-5. https://doi.org/10.1080/14786419.2018.1552697
ACQUAVIVA R., MENICHINI F., RAGUSA S., GENOVESE C., AMODEO A., TUNDIS R., LOIZZO M. R., IAUK L. (2013). ANTIMICROBIAL AND ANTIOXIDANT PROPERTIES OF BETULA AETNENSIS RAFIN. (BETULACEAE) LEAVES EXTRACT. NATURAL PRODUCT RESEARCH, 27(4-5):475-479. ISSN: 1478-6419.
Author Response
- Introduction Line 62-63:
The AA stated: “Nowadays, a number of plant-based agents are clinically used in cancer treatment proving the goodness of these type of products”; better to use the term “support to anticancer therapy”, and to indicate more specific references, such as ACQUAVIVA R, SORRENTI V, SANTANGELO R, CARDILE V, TOMASELLO B, MALFA G, VANELLA L, AMODEO A, GENOVESE C, MASTROJENI S, PUGLIESE M, RAGUSA M, DI GIACOMO C (2016). EFFECTS OF AN EXTRACT OF CELTIS AETNENSIS (TORNAB.) STROBL TWIGS ON HUMAN COLON CANCER CELL CULTURES. ONCOLOGY REPORTS, 36(4):2298-2304. ISSN: 1791-2431, DOI: http://dx.doi.org/10.3892/or.2016.5035
- Done as suggested.
- Add : The activity is confirmed on extracts having the same chemical composition by others AA. ( insert references: ACQUAVIVA R, D’ANGELI F, MALFA GA, RONSISVALLE S, GAROZZO A, STIVALA A, RAGUSA S, NICOLOSI D, SALMERI M, GENOVESE C. (2019) ANTIBACTERIAL AND ANTI-BIOFILM ACTIVITIES OF WALNUT PELLICLE EXTRACT (JUGLANS REGIA L.) AGAINST COAGULASE-NEGATIVE STAPHYLOCOCCI. NATURAL PRODUCT RESEARCH, 1-6. DOI: 10.1080/14786419.2019.1650352
- The authors consider that the statements made in that paragraph are sufficiently contrasted by the references given. You propose to incorporate a reference of a work whose main objective is the determination of antibacterial and anti-biofilm of a dried fruit that is not the same as we used in our study.
We greatly appreciate this work, it seems to us that the authors made a great contribution to science with it and we are sure that we will use it as a reference in future studies, since it seems to us a very promising line of research.
- The works n. 25 and 26 are very old, replace:
- The works mentioned are old because they are the original works. Being old references does not make them less important. They are the authors who discovered the techniques we use today for this type of analysis. We believe that it would not be appropriate not to cite them since it would be a depletion of their work.
Reviewer 2 Report
The study deals with influence of water stress (hydroSOS approach) on possible improvement of various biological activities related to antioxidant activity of different rootstocks of pistachio nuts. The study seems scientifically sound, but still I have few comments.
Section materials and methods (line 252): I do not really understand why the extract were prepared using MeOH with added ascorbic acid. A 1% is quite a concentrated solution. In my opinion, since ascorbate is a strong antioxidant in vitro, it would definitely affect the final antioxidant properties of the extracts in a positive manner. Could the authors explain why they did so?
Furthermore, I also do not get why the authors used cytotoxic (anticancer) assay. I cannot imagine how water stress would be associated with subsequent production of potentially toxic compounds in a plant, which is otherwise quite drought-resistant. Or, is there any indication that eating pistachios from arid zones is associated with higher incidence of toxicity/anticancer activity? I would welcome some explanation why authors chose this method, in either section methodology or results and discussion.
Lastly, I suggest that the discussion could be more developed, e.g. the pistachio extracts showed selective anticancer activity: what compounds are responsible for this activity? By what mechanism is this selectivity achieved? Is it safe to consume pistachios with anticancer activity? Would all observed in vitro activities (antioxidant, antimutagenic, cytoprotective) likely be also working in vivo? In connection to that, there have been some indications quite recently that cellular models of oxidative stress involving use of hydrogen peroxide might not always behave in a linear fashion and thus these methods have some limitations1,2. This fact might be involved in discussion as well.
After the above-mentioned issues are addressed, I think that the manuscript is ready to be published in Molecules.
References:
Kaczara, P., Sarna, T. & Burke, J. M. Dynamics of H2O2 availability to ARPE-19 cultures in models of oxidative stress. Free Radic. Biol. Med. 48, 1064–1070 (2010). Halliwell, B. Free radicals and antioxidants: Updating a personal view. Nutr. Rev. 70, 257–265 (2012).Author Response
- The study deals with influence of water stress (hydroSOS approach) on possible improvement of various biological activities related to antioxidant activity of different rootstocks of pistachio nuts. The study seems scientifically sound, but still I have few comments.
Thank you very much for your comments. The authors greatly appreciate the effort you have made by reviewing this article. We are completely sure that with your help we have managed to improve its quality.
- Section materials and methods (line 252): I do not really understand why the extract were prepared using MeOH with added ascorbic acid. A 1% is quite a concentrated solution. In my opinion, since ascorbate is a strong antioxidant in vitro, it would definitely affect the final antioxidant properties of the extracts in a positive manner. Could the authors explain why they did so?
We use this type of extractants based on the existing bibliography and our previous experience, which has shown us that this composition is the one that is capable of extracting a greater number of polyphenols. Prior to this study, we made a characterization of the polyphenols present in our pistachio samples. These results were already published:
Noguera-Artiaga, L., Pérez-López, D., Burgos-Hernández, A., Wojdyło, A., Carbonell-Barrachina, Á.A. Phenolic and triterpenoid composition and inhibition of α-amylase of pistachio kernels (Pistacia vera L.) as affected by rootstock and irrigation treatment (2018) Food Chemistry, 261, pp. 240-245.In this study, we follow the methodology used to compare the data and establish a relationship with them.
This type of extraction has already been previously used by other authors, for example:
- Furthermore, I also do not get why the authors used cytotoxic (anticancer) assay. I cannot imagine how water stress would be associated with subsequent production of potentially toxic compounds in a plant, which is otherwise quite drought-resistant. Or, is there any indication that eating pistachios from arid zones is associated with higher incidence of toxicity/anticancer activity? I would welcome some explanation why authors chose this method, in either section methodology or results and discussion.
It is well known that when a plant is stressed under controlled conditions, it allocates CO2 to the formation of secondary metabolites in order to counteract the free radicals formed as a result of that stress. That is the main hypothesis on which we rely on when carrying out this study. We believe that controlled water stress, apart from achieving considerable water savings during cultivation, would increase the concentration of bioactive compounds in the plant and that they had some cytotoxic power against the cells studied. There is much literature in this regard that mentions that some polyphenols have cytotoxic power against cancer cells, while they do not have it in the case of normal cells. Prior to this study, we analyzed the effect that these factors had on the polyphenolic composition of the fruits and observed that mild water stress (under the same conditions described in this article) increased the concentration of some polyphenols, such as polymeric procyanidins:
Noguera-Artiaga, L., Pérez-López, D., Burgos-Hernández, A., Wojdyło, A., Carbonell-Barrachina, Á.A. Phenolic and triterpenoid composition and inhibition of α-amylase of pistachio kernels (Pistacia vera L.) as affected by rootstock and irrigation treatment (2018) Food Chemistry, 261, pp. 240-245.This information is included and referenced in the results and discussion section. Please see lines 115-118.
- Lastly, I suggest that the discussion could be more developed, e.g. the pistachio extracts showed selective anticancer activity: what compounds are responsible for this activity? By what mechanism is this selectivity achieved? Is it safe to consume pistachios with anticancer activity? Would all observed in vitro activities (antioxidant, antimutagenic, cytoprotective) likely be also working in vivo?
We believe that the compounds responsible for this effect may be the triterpenoids present in pistachio. This thought is in line with results obtained by other authors, as we have reflected in lines 177-182.
When we refer to cytotoxicity, we refer to the ability to inhibit or destroy cancer cells. As you well know, cancer cells are unregulated cells. They do not have an internal regulation mechanism and are more or less unbalanced. They generate a large amount of free radicals and accumulate them. When a new substance with great antioxidant power enters, as is the case of triterpenoids, polymeric procyanidins or other polyphenols are likely to be able to further de-structure this disorder and cause cell death or a reduction in their multiplication. This type of effect is not observed in normal cells since they are able to restore their functioning perfectly. Therefore, it is safe to eat pistachios and not because we say so, but because it is a food that has been consumed for millennia.
The effects observed in vitro are not able to ensure that they will be reproduced in vivo. It is necessary to do studies and contrast the information in order to have an idea about it. On the other hand, we believe that the effect observed in vitro will be diminished in vivo since a large part of the compounds to which we attributed this effect (triterpenoids or polyphenols) oxidize rapidly during the digestion process.
But, in our opinion, what is more important is that it has been observed in this study as well as in others referenced in the discussion section that there are compounds present in pistachio that may have cytotoxic capacity against carcinogenic cell lines. Currently, we work in the isolation of these compounds in order to carry out individual studies.
- In connection to that, there have been some indications quite recently that cellular models of oxidative stress involving use of hydrogen peroxide might not always behave in a linear fashion and thus these methods have some limitations1,2. This fact might be involved in discussion as well.
Thank you for this comment. We added this information. Please, see lines 219-222. We are sure that this paper has improved greatly with your comments. Thank you.
Reviewer 3 Report
other papers have beeen publishen in pistachhio nuts, howerver, in this work stressed cultivating conditions in pistachio cultivars P. atlantica, P. integerrima, and P. terebinthus) such as regulated deficit irrigation (RDI) created a plant response that increased the production of secondary metabolites, and increased the antioxidant (ABTS, FRAP, and DPPH), antimutagenic (Ames test), and cytotoxicity (MTT assay in 5 human cell lines) activities. the paper is sound, and results are promising. only commment is to check small mistakes in english.
Author Response
- Other papers have beeen publishen in pistachhio nuts, howerver, in this work stressed cultivating conditions in pistachio cultivars P. atlantica, P. integerrima, and P. terebinthus) such as regulated deficit irrigation (RDI) created a plant response that increased the production of secondary metabolites, and increased the antioxidant (ABTS, FRAP, and DPPH), antimutagenic (Ames test), and cytotoxicity (MTT assay in 5 human cell lines) activities. the paper is sound, and results are promising. only commment is to check small mistakes in english.
- Thank you very much for your comment. We appreciate the time you have spent to review this article and we appreciate your feedback.
We have reviewed the text and corrected some small formatting errors.